Historical and current distribution ranges and loss of mega-herbivores and carnivores of Asia

http://orcid.org/0000-0002-2432-7732 Mahmood Tariq 1 2
http://orcid.org/0000-0002-6656-8016 Vu Tuong Thuy 2 3
http://orcid.org/0000-0002-4657-4216 Campos-Arceiz Ahimsa 2 4
http://orcid.org/0000-0001-9313-0637 Akrim Faraz 1 5 farazakrim@uaar.edu.pk
http://orcid.org/0000-0003-2198-0047 Andleeb Shaista 6
Farooq Muhammad 1
Hamid Abdul 1
Munawar Nadeem 1
Waseem Muhammad 1
Hussain Abid 1
Fatima Hira 1
Khan Muhammad Raza 1
http://orcid.org/0000-0001-5693-0985 Mahmood Sajid 7
1 Department of Wildlife Management, PMAS-Arid Agriculture University Rawalpindi , Rawalpindi, Punjab , Pakistan
2 School of Geography, University of Nottingham Malaysia Campus , Semenyih, Selangor , Malaysia
3 Faculty of Engineering and Science, Curtin University Malaysia , Miri, Sarawak , Malaysia
4 Southeast Asia Biodiversity Research Institute, Xishuangbanna Tropical Botanical Garden, Chinese Academy of Sciences , Xishuangbanna , China
5 Department of Zoology, University of Kotli , Azad Jammu and Kashmir , Pakistan
6 School of Resources and Environmental Engineering, Wuhan University of Technology , Wuhan, Hubei Province , China
7 Department of Zoology, Hazara University , Mansehra, Khyber Pakhtunkhwa , Pakistan
Nganvongpanit Korakot
Electronic publication date: 2021 Feb 16
Publication date: 2021
Volume: 9
Electronic Location ID: e10738
Received 2020 Aug 7; Accepted 2020 Dec 18
Copyright: © 2021 Mahmood et al.
Copyright year: 2021
Copyright holder: Mahmood et al.
License: This is an open access article distributed under the terms of the Creative Commons Attribution License, which permits unrestricted use, distribution, reproduction and adaptation in any medium and for any purpose provided that it is properly attributed. For attribution, the original author(s), title, publication source (PeerJ) and either DOI or URL of the article must be cited.
License URL: https://creativecommons.org/licenses/by/4.0/

Keywords: Large herbivore, Large carnivores, Mega-defaunation, Mega-gardners, Distribution range

Funding: The authors received no funding for this work.

==============================
Ecosystem functioning is dependent a lot on large mammals, which are, however, vulnerable and facing extinction risks due to human impacts mainly. Megafauna of Asia has been declining for a long, not only in numbers but also in their distribution ranges. In the current study, we collected information on past and current occurrence and distribution records of Asia’s megafauna species. We reconstructed the historical distribution ranges of the six herbivores and four carnivores for comparison with their present ranges, to quantify spatially explicit levels of mega-defaunation. Results revealed that historically the selected megafauna species were more widely distributed than at current. Severe range contraction was observed for the Asiatic lion, three rhino species, Asian elephant, tigers, and tapirs. Defaunation maps generated have revealed the vanishing of megafauna from parts of the East, Southeast, and Southwest Asia, even some protected Areas losing up to eight out of ten megafaunal species. These defaunation maps can help develop future conservation policies, to save the remaining distribution ranges of large mammals.

Introduction

Megafauna species—the largest vertebrates (Hansen & Galetti, 2009)—were once present in most terrestrial ecosystems (Smith et al., 2010), where they play key roles in the top-down regulation of ecosystem processes. Megafaunal loss results in trophic cascades with large-scale impacts (Estes et al., 2011). The regional loss of megaherbivores, for example, has been linked to changes in nutrient biogeochemistry (Doughty, Wolf & Malhi, 2013), climate (Doughty, Wolf & Field, 2010), and seed dispersal processes (Janzen & Martin, 1982; Campos-Arceiz & Blake, 2011), among others. The loss of apex predators such as wolves in North America has also been linked to changes in vegetation, pollination, and even local geomorphology (Beschta & Ripple, 2012). For these reasons, megafauna is often described as a keystone (Owen-Smith, 1989), strongly interacting (Soulé et al., 2005), or ecosystem engineering (Campos-Arceiz, 2009) species. But, due to their high resource requirements and tendency to be k-strategists, megafauna species also tend to be very sensitive to human impacts (mainly hunting and habitat loss) and are particularly prone to local and global extirpation (Woodroffe & Ginsberg, 1998; Milner-Gulland et al., 2003; Cardillo et al., 2005).

There is indeed a long history of megafauna extirpation by humans. The first global wave of human-driven megafaunal extinctions occurred during the Quaternary period, approximately 50,000 to 10,000 years ago (Bonnichsen, 1988; Barnosky, 2008; Smith et al., 2018). This Quaternary Mass Extinction (QME) event was eliminated without replacement about two-thirds of all mammalian genera and one-half of all species having a body mass greater than 44 kg (Barnosky, 2008). Since the Middle Pleistocene (781–126 thousand years ago) the majority of terrestrial ecosystems outside Africa have lost megafaunal vertebrates greater than 44 kg body mass (Corlett, 2013). However, the QME event had differential effects across continents—whereby the Americas and Australia lost almost all their megafauna and Africa suffered no major losses, Asia suffered a mild effect, probably due to the long presence of Homo erectus (since ~ 1.6 mya) in the region.

The global anthropogenic changes caused by the growth of the human population and geographical distribution, coupled with increased technological capacity in the past few centuries have resulted in another wave of megafaunal decline. Factors including, hunting, habitat fragmentation, habitat loss, human-wildlife conflict, and various anthropogenic activities have accelerated megafauna loss rates.

Understanding and quantification of historic ranges of threatened megafauna is a prerequisite for the development of conservation and restoration policies (Laliberte & Ripple, 2004). Laliberte & Ripple (2004) assessed changes in the distribution range of 43 North American carnivores and ungulates since the 19th century and reported a loss of species richness and range contraction of >20% in about one-third of the species. Ceballos & Ehrlich (2002) reported that among 173 threatened mammals from six different continents, have lost greater than 50% of their distribution ranges during the past two centuries. Globally, it is estimated that <21% of the earth’s terrestrial surface has an intact assemblage of large mammals (>20 kg) (Morrison et al., 2007). The Indomalayan region, having a great diversity of large mammals (Soberón & Ceballos, 2011; Ripple et al., 2016) has faced mammal decline (Ceballos & Ehrlich, 2002; Sodhi et al., 2010; Ripple et al., 2017), and has only maintained intact large-mammal assemblage of 1% on its terrestrial area (Morrison et al., 2007). Earlier studies have documented range contractions over time ranging from decades (Worm & Tittensor, 2011) to a few centuries (Laliberte & Ripple, 2004; Ceballos & Ehrlich, 2006; Morrison et al., 2007). The decline in Megafauna has been taking place in parts of tropical Asia, for several millennia (Elvin, 2004).

The purpose of this study was to record the historical distribution of Asia’s megafauna over a period of approximately 10,000 years to identify the level of “mega-defaunation” across the region and identify priority areas for conservation action. Specifically, our objectives were to (1) collect data on the historical distribution ranges of selected Asian megafaunal species; (2) compare their historical and current distribution ranges; and (3) quantify megafaunal species loss in natural habitats, represented here by the network of Protected Areas (PAs) in the region. Our study provides spatially explicit information on “mega-defaunation” levels that can be used in the design of conservation policies, particularly for the restoration of megafaunal populations and their ecological function.

Materials and Methods

Geographical and temporal scope

The geographical scope of our analyses was “Asia” in senso lato. Specifically, we considered mainland Asia up to approximately 35° west and 40° north (we are aware that this is further north than standard tropical limits); for example, Corlett (2013) and the islands of Sri Lanka, Sumatra, Borneo, Java, Hainan, and Taiwan (Fig. 1A). We consider “historical distribution” as the natural occurrence of a species anytime in the past ~10,000 years.

Figure 1 Flow diagram of database searching and screening for meta-analysis.

Species considered

We analyzed variations in the distribution range of ten large mammal species, including six herbivores—the Asian elephant (Elephas maximus), Indian or greater one-horned rhinoceros (Rhinoceros unicornis), Javan or lesser one-horned rhinoceros (Rhinoceros sondaicus), Sumatran or two-horned Asian rhinoceros (Dicerorhinus sumatrensis), gaur (Bos gaurus), and Malayan tapir (Tapirus indicus)—and four carnivores—tiger (Panthera tigris), Asiatic lion (Panthera leo persica), common leopard (Panthera pardus), and clouded leopard (Neofelis nebulosa; Table 1). These species are a non-exhaustive representation of the largest terrestrial mammals in tropical Asia and their selection was based on their role in the ecosystem including both herbivore and carnivores since both groups have a considerable impact on the ecosystem.

Table 1 Comparison of historical and current distribution ranges of the ten selected mega-faunal species and percent range reduction that has occurred through history.

	Animal species	Home range size (km2)	Maximum elevation above sea level (m)	Historical distribution (km2)	Current distribution (km2)	Range reduction (%)	
1	Asian Elephant	320	3,000	1.2 × 107	6.9 × 105	95.1	
2	Gaur	–	–	5.6 × 106	1.5 × 106	72.0	
3	Indian Rhinos	–	–	2.5 × 106	4,193.2	99.8	
4	Javan Rhinos	–	2,000	8.5 × 106	252.69	100	
5	Sumatran Rhinos	50	–	7.5 × 106	8,879.3	99.9	
6	Malay Tapir	–	–	4.5 × 106	91,947	98.0	
7	Asiatic Tiger	4,000	4,360	2.3 × 107	1.6 x 106	92.9	
8	Asiatic Lion	–	–	1.1 × 107	*1,412	100	
9	Common Leopard	78	5,300	3.2 × 10 7	8.4 × 106	73.7	
10	Clouded leopard	–	–	6.3 × 106	2.2 × 106	64.0	

Data on historical and current distribution

The location data on historical distribution/occurrence of the ten target megafaunal species were collected from published and unpublished literature, our sources included journal articles, books, research thesis, newspaper articles, and personal communications with reputable scientists. Historic location data on the distribution of focal species were collected as described by Mahmood et al. (2019). Data on ecological parameters were also collected including vegetation type, altitude, etc., when available. The articles or records having weak evidence and location data were excluded from the analysis to remove bias and only those sources and records were considered having accurate location data of target species (Mahmood et al., 2019).

Location data were then imported into google earth software and saved as KML (keyhole Markup Language) files which were then converted into Shapefiles in QGIS (Quantum Geographic Information System) software (Mahmood et al., 2019). The output was a vector layer having a known historical distribution of focal species (blue dots; Fig. 1). Using toggle editor in QGIS software we filled gaps in distribution ranges of species based on previously existing historical maps and ecological factors. During filling gaps in distribution maps the inclusion criteria were if we lacked information related to the historical presence/distribution of a species in an area surrounded by known historical presence and there was no apparent ecological barrier or difference with the surroundings, we considered it as part of the historical distribution range (Mahmood et al., 2019).

The data on the current distribution of nine of the target species (all but Asiatic lions) were downloaded from the website of IUCN’s Red List of Threatened Species (http://www.iucnredlist.org/technical-documents/spatial-data) as a shapefile document, which is the most reliable source documenting the current distribution of focal species (Fig. 1).

Map of protected areas

We used the World Database on Protected Areas (https://www.protectedplanet.net/) (UNEP-WCMC, 2015) to map protected areas (PAs) occurring in our study area. The original dataset included more than 8,350 PAs in our area of concern. Of these, we decided to use only terrestrial PAs larger than 20 km2 of size. After excluding small PAs and the category “marine protected areas”, our dataset retained 4,773 PAs, ranging from the 21.1 km2 of Ampang Catchment Forest Reserve (Malaysia) to Bukit Batutenobang NP (8,830 km2) in Borneo.

Defaunation analyses

The historical and current distribution ranges of each species were mapped in the form of dots (past distribution points) and polygons (current distributions as per IUCN data) and a comparison was then made between the historical vs. current occurrence records for each mega-faunal species under study, to highlight the areas that suffered mega-defaunation since historical times.

For each species, we constructed past and current distribution range and created a new layer including protected areas classified based on (i) protected areas where focal species were never present, (ii) protected areas where focal species were present in the past but now are extinct, (iii) and protected area where focal species are still present. All analysis was conducted using QGIS software as described by Mahmood et al. (2019).

Finally, we summarized the information in four Index maps showing the total number of megafaunal species (from the 10 included in our study) historically and currently (2008) present in the large PAs of tropical Asia; and the absolute and relative loss of megafaunal species per PA. The percent of defaunation in each protected area was simply calculated using: D×100H

where, D represents the difference in numbers of megafaunal species in a PA between historical and current times, and H represents the number of species present in the PA in history. Besides, the total number and size of PAs of Asia were also calculated to assess the percent of PAs that have lost particular megafaunal species during the course of history.

Results

Initially, we downloaded 2,832 occurrence records of megafauna in different forms. After removing duplicate records, we were left with 2,450 documents which were further screened and 903 documents were excluded based on weak evidence, and incomplete information. The remaining 1,547 articles were further assessed and 237 further articles were removed based on weak evidence and location data to remove bias and the remaining 1,310 articles were used for quantitative synthesis and meta-analysis (Fig. 1).

Individual species

Asian elephant

Asian elephants, in historical times, occurred up to Turkey through west Asia along the Iranian coast; in the Indian subcontinent, China, and spread into Southeast Asia up to Sumatra, Borneo—but see (Cranbrook, Payne & Leh, 2008)—and Java (Fig. 2). Their historical distribution was reconstructed using 458 points (see OSM for details). The historical distribution records indicate that Asian elephants occurred from northeast China towards the south through Thailand, Cambodia, Vietnam, Malaysia, Sumatra, and Java, while towards the west, it ranged through Bangladesh, Nepal, India, Pakistan, Iran, and Afghanistan up to Turkey. Whereas currently (Fig. 2) its IUCN distribution shows the species occurs in a much-reduced range including Indonesia, Malaysia through Thailand, Bangladesh, China, Nepal, India, and Sri Lanka over an area of approximately 6.9 million km2 (Table 1). In the past, Asian elephants were distributed among approximately 2206 Protected Areas of Asia measuring approximately 1.2 million km2, whereas at present they occur only in 310 Protected Areas with a total size of approximately 300,000 km2 (Fig. 2; Table 2). Accordingly, Asian elephants are ecologically missing in 75.2% of the PAs (or 83.4% of the PA area) that historically hosted them.

Figure 2 Historical and current distribution ranges of six herbivore megafauna’ species in Protected and non-Protected Areas in Asia.

(A) Asian elephant, (B) Gaur, (C) Indian rhino, (D) Javan rhino, (E) Sumatran rhino, and (F) Malay tapir.

Table 2 Size (area) of protected areas that contained selected mega-faunal species in history and at current times and their percent reduction.

Animal species	Historical distribution	Current distribution	Defaunation	
N PAs	Area (km2)	N PAs	Area (km2)	% PAs (N)	% PA’s (Area)	
Mega-herbivores	
Asian elephant	2,206	1,222,877	310	303,300	83.4	75.2	
Gaur	1,818	793,958	484	343,468	73.4	56.1	
Indian rhinos	465	2,645,013	10	4,193	95.7	99.8	
Javan rhinos	2,909	1,156,427	2	112	99.9	100	
Sumatran rhinos	2,974	1,016,761	21	15,617	99.4	98.5	
Malay tapir	1,987	643,009	96	56,666	95.2	91.2	
Mega-Carnivores	
Asiatic tiger	4,014	2,211,656	132	463,822	96.7	79.0	
Asiatic lion	892	304,802	1	1,412	99.9	99.5	
common leopard	4,864	2,249,670	451	1,120,815	90.7	50.2	
clouded leopard	2,234	896,598	813	409,605	63.6	54.3	
Notes:

* Total Area covered over by PA’s of Asia = 7,418,073.04103 km2.

Total numbers of PA’s of Asia = 8,398.

Gaur

The historical range reconstructed in the current study based on 134 points has revealed that historically, gaurs were distributed throughout mainland South and Southeast Asia, from India and Sri Lanka up to the Malay Peninsula (Fig. 2). But gaurs are now extirpated from Sri Lanka and in other countries occur in a scattered distribution but still covering an area of approximately 1.5 million km2, as per IUCN 2008 distribution maps as against 5.6 million km2 (Table 1; Fig. 2) in the past. In terms of protected areas, the species occurred historically in 1,818 of the current PAs, whereas at present it occurs in 484 PAs (Table 2; Fig. 2). In terms of protected area size, gaurs have disappeared from approximately 56.1% of PA’s that harbored them in the past (Fig. 2; Table 2).

Indian rhino

We reconstructed the historical distribution of Indian rhinos using 36 occurrence points mentioned in the previously published literature, whereby the rhino species ranged in the northern part of the Indian subcontinent, from Pakistan to the India-Burma border, through Nepal, Bangladesh, Bhutan, and southern China (Fig. 2). Indian rhinos used to occur in what now would be 465 Protected Areas (Table 2). Currently, however, Indian rhinos are restricted to a few small populations in Nepal and India, occupying an area of 4,193 km2, occurring in 10 PAs (Table 2). Based on their historical distribution, Indian rhinos, are ecologically missing from approximately 99.8% of their historical range and from 95.7% of PAs where they once occurred (Tables 1 and 2; Fig. 2).

Javan rhino

We used a total of 71 location points mentioned in the previously published literature to reconstruct the historical distribution of Javan rhinos, which revealed that they used to occur from Java and Sumatra up to India through the Malay peninsula, Thailand, Cambodia, Laos PDR, Vietnam, Southern China, Myanmar, Bangladesh, Bhutan, and Nepal. (Fig. 2). By 2008, however, Javan rhinos’ range was restricted to Cat Tien National Park in Vietnam (a population that sadly went extinct in 2010; Brook et al. (2012), and Java’s Ujung Kulon Peninsula). The species historical range overlapped with 2909 PAs, while by 2008 they were found just in the two PAs as mentioned earlier (Tables 1 and 2) sadly only one now (Campos-Arceiz & Teckwyn, 2019). The Javan rhinos have lost 98.7% of their historical area that now falls under PAs (Fig. 2; Tables 1 and 2).

Sumatran rhino

Sumatran rhinos’ historical distribution was reconstructed using 243 location points mentioned in the literature. According to these records, the species ranged from Sumatra and Borneo up to the Himalayan foothills in Bhutan through the Malay Peninsula, Thailand, Cambodia, Lao PDR, Vietnam, southern China, Myanmar, and northeastern India (Fig. 2). Historically, Sumatran rhinos occurred in what is now 2,974 PAs, occupying an area of about 1.0 million km2 (PA’s), while in 2008 they were estimated to occur in just 21 PAs having a total area of 15,617 km2 (Fig. 2; Table 2; unfortunately their range has further reduced and is now limited to four populations in Sumatra and Kalimantan). By the year 2008, their estimated distribution range was approximately 8,880 km2 (as per IUCN maps). Based on their historical distribution, Sumatran rhinos have disappeared from 98.5% of the PAs where they historically occurred (Fig. 2; Tables 1 and 2).

Asian/Malay tapir

We reconstructed the species historical range using a total of 29 points, according to which, Asian tapirs historically occurred in China, southern Cambodia, southern Vietnam, Lao PDR, Thailand, Myanmar and India, and the Islands of Sumatra, and Java (Fig. 2). At present, however, Asian tapirs are extinct in China, Cambodia, Vietnam, Lao PDR, and India, and the size of their current distribution range (as per IUCN’s 2008 map) is only 91,947 km2 (Table 1 and 2). Asian tapirs had historically occurred in 1987 current PAs, while at present the species occurs in only 96 PAs (Tables 1 and 2). Hence, Asian tapirs have been lost in approximately 91.2% of the protected area where they had occurred in history (Fig. 2; Table 2).

Tiger

Historically, tigers were widely distributed across Asia, from Turkey in the west through South and Southeast Asia up to the eastern coasts of Russia, in the form of nine subspecies of which only six survive today. Tigers’ historical range was reconstructed using a total of 193 location points mentioned in published literature (Fig. 3). However, by 2008 the species was restricted to 13 countries including Bangladesh, Bhutan, Cambodia, China, India, Indonesia, Lao PDR, Malaysia, Myanmar, Nepal, Russia, Thailand, and Viet Nam, covering a total distribution range of approximately 1.6 million km2 as per IUCN recent data (Table 1). Historically, tigers were distributed in 4014 PAs covering an area of approximately 2.2 million km2 (as PAs). By 2008, tigers occurred in only 132 PAs with a total PAs size of approximately 0.46 million km2 (Table 2). Therefore, approximately 79 % of the PAs that historically hosted tigers have lost them (Table 2; Fig. 3).

Figure 3 Historical and current distribution ranges of four carnivore megafauna’ species in Protected and non-Protected Areas in Asia.

(A) Asiatic tiger, (B) Asiatic lion, (C) Common leopard, (D) Clouded leopard, and (E) Sunda clouded leopard.

Asiatic lion

Fossil records and historical accounts show that in historical times, Asiatic lions ranged from southwest Asia (Iraq, Iran, Pakistan, and India) through eastern India up to North Africa, Central Asia (Ukraine, Armenia, Azerbaijan), and Europe (Italy, Greece, Bulgaria, Macedonia, Hungary, Turkey, Russia) (Fig. 3). We used 44 total numbers of location points extracted from published literature to reconstruct this historical distribution in Asia (Table 1). The distribution/location points of Asiatic lions that fall outside of Asia, especially North Africa and Europe, were not included in the current study and analysis. By 2008, this apex carnivore had become confined to a single place: the Gir forest of Gujarat, India, occupying a range size of just 1,412 km2 (Table 2). In the past, this top carnivore species occurred in 892 PAs, covering a size of approximately 300,000 km2 (Table 2). Hence, Asiatic lions have disappeared from approximately 99.5% of the PAs that historically had hosted them (Tables 1 and 2; Fig. 3).

Common leopard

Common leopards historically had a much broader range, occurring in the form of nine subspecies from Turkey into Southwest Asia (including Iran, Afghanistan, Pakistan), Nepal, Bhutan, India, Sri Lanka, Bangladesh, Myanmar, China, North and South Korea, Thailand, Laos, Vietnam, Cambodia, Malaysia and the island of Java; they also occurred in Oman, UAE, Central Asian states and Europe including Turkmenistan, Azerbaijan, Uzbekistan, Georgia, Armenia, and Russia (Fig. 3). While still widely distributed, common leopards’ range has been constrained to ca 8.4 million km2 from their historical range (Table 1). Common leopards occurred historically in 4,864 PAs, covering a cumulative protected area of approximately 2.25 million km2 (Table 2). By 2008 they occurred in 451 PAs, having a total size of approximately 1.12 million km2. This is, approximately 90% of the PAs and about 50.2% PA area that historically had leopards have lost this top predator (Table 2; Fig. 3).

Clouded leopard

More recently, the clouded leopard has been split up into two distinct species based on genetic analysis; the clouded leopard (Neofelis nebulosa) and the Sunda clouded leopard (Neofelis diardi). Based on a total of 73 location points mentioned in the published literature about occurrence of N. nebulosa in the past, we reconstructed its historical distribution range (Fig. 3). The species N. nebulosa had a wider distribution in history than the current one—it ranged from India and Nepal up to Peninsular Malaysia through Bangladesh, and China (south of Yangtze) but today, its distribution range has been restricted, and according to IUCN estimates, it covers an area of approximately 2.2 million km2 (Table 1; Fig. 3). The clouded leopard occurred in what is now 2,234 PAs, covering a total area of approximately 0.89 million km2. Today, clouded leopards occur in 813 PAs which cover an area of approximately 400,000 km2 (Table 2). In terms of numbers of PAs, clouded leopards had disappeared from approximately 63.6% of the PAs and 54.3% of the PA area that they historically occupied (Table 2; Fig.3). The other species, the Sunda clouded leopard was distributed on Borneo and Sumatra in the past and has not shown much range changes.

General defaunation patterns

In whole Asia, a total of 8380 protected areas of various categories occur according to the “world database on protected areas (WDPA)”, including marine PAs that were not included in our analysis. Also, we excluded 3,577 PAs from the current analysis because of their small size (being less than 20 km2). So the total numbers of PA’s analyzed in the current study were 4,773, ranging from 20 km2 to PAs as large as Kerinci Seblat NP (13,750 km2), Sunderban South WLS (36,970 km2), Cholistan Game Reserve (20,326 km2), and Touran NP (14,706 km2; Tables 1 and 2).

Historically, a higher number of megafaunal species were present in SE than in SW Asia (Figs. 4A–4D). For example, as many as eight out of our 10 studied species were expected to have occurred sympatrically in SE Asian PAs such as Taman Nagara NP (Peninsular Malaysia), Tonle Sap Biosphere Reserve (Vietnam), and Nam Chuane Conservation area (Lao PDR). The PAs rich in megafauna outside SE Asia include Royal Bardia NP (Nepal), where also eight megafauna species occurred; Yangzie Nature Reserve (China), with seven species. In South Asia, smaller numbers of megafaunal species co-existed historically (Fig. 4)—for example, a maximum of six species occurred in Simlipal NP and Kaimur sanctuary (India). Further towards SW Asia, in Iran and Iraq, the number of sympatric species decreases considerably: for example, three species in Bahukalat and two in Kavir NP Iran (Table 3).

Figure 4 Index maps of ten selected megafauna species.

(A) Showing total numbers of megafauna species in PA’s of Asia in history, (B) total numbers of megafauna species at current time, (C) defaunation index map showing the difference in numbers of megafauna species in PA’s of Asia between historic and current times, and (D) index map showing percent loss of megafauna in PA’s of Asia since history.

Table 3 Details of the PA’s with their size, location, and the numbers of megafauna species they had in history and at current and the percent of defaunation that has occurred in these PA’s.

Sr. No.	Name of PA	Size
(km2)	Country	No. of species in history	No. of species at current	% Defaunation	
1	Taman Negara NP	4,524.54	Malaysia	8	7	12.5	
2	Tonle Sap Biosphere	3,222.69	Cambodia	8	0	100	
3	Royal Chitwan NP	750	Nepal	8	6	25	
4	Gunung Leuser NP	7,926.75	Indonesia	5	4	20	
5	Kerinci Seblat NP	1,3750	Indonesia	6	5	16	
6	Bukit Batutenobang NP	8,830	Borneo	4	0	100	
7	Margalla Hills NP	173.86	Pakistan	4	0	100	
8	Wu Ling Yuan NP	264	China	6	0	100	
9	Yangzie Nature Res	433.33	China	7	1	85	
10	Taungup Pass	–	Myanmar	6	2	66	
11	Sunderban South WLS	3,6970	India	3	0	100	
12	Gir Forest NP	258.71	India	3	2	33	
13	Yala NP	289.04	Sri Lanka	3	1	66	
14	Walpattu NP	549.53	Sri Lanka	3	1	66	
15	Wasgomuva	369.48	Sri Lanka	3	1	66	
16	Touraun NP	14,706.4	Iran	2	0	100	
17	Cholistan Game Reserve	20,326.67	Pakistan	5	0	100	
18	Lang Tang NP	1710	Nepal	7	1	85	
19	Ben En NP	166.34	Vietnam	7	0	100	
20	Vo Doi NP	33.94	Vietnam	5	0	100	
21	Bali Barat	190.03	Indonesia	3	0	100	
22	Ujong Kulon NP	1,229.56	Indonesia	4	2	50	
23	Kota Kinabalu NP	753.7	Saba Malaysia	4	1	75	
24	Khao Yai NP	2,165.55	Thailand	8	3	62	
25	Trishna WLS	194.7	India	8	0	100	
26	Wolong NR	2,000	China	5	1	80	
27	Barail WLS	300	India	7	0	100	
28	Rema Kalenga WLS	–	Bangladesh	8	0	100	
29	Belum WLR	2072	Malaysia	8	6	25	
30	Cat Tien NP	738.78	Vietnam	7	4	42	
31	Royal Bardia NP	968	Nepal	8	4	50	
32	Kaziranga NP	849.79	India	7	4	42	
33	Pegu Yomas NP	1,463.35	Myanmar	7	3	57	
34	Namdapha NP	1,807.82	India	7	3	57	
35	Manas NP	391	India	8	3	62	

The current scenario shows drastic changes in the distribution of Asian megafaunal species (Figs. 4B–4D)—some PAs that should be rich in megafauna have suffered total defaunation. Notable examples include Tonle Sap Biosphere Reserve (Vietnam), which has lost all of its original eight species; Dahongshanyinxing and Poyanghuhouniao Nature Reserves (China), which have lost their seven species; and Wu Ling Yuan (China), which has lost all six species. Noradehi Sanctuary (India) has lost five species; Chumbi Surla WLS, Thal Game Reserve, Nara desert WLS, and Diljabba-Domeli Game Reserve (Pakistan) retain none of the five historical megafauna species; and Dareh Anjir and Neibaz Wildlife Refuge (Iran) have lost all of their original three historical megafauna species (Fig. 4B). Cases of PAs that have lost most of their megafaunal species are much more abundant (Table 3).

Importantly, none of the PAs in our study have been successful in retaining all their historical megafaunal species, although a few PAs have retained more than 70% such as Taman Negara NP (Malaysia, retained 6 out of eight species), Kerinci Seblat NP (Indonesia, retained five out of six), and Gunung Leusur NP (Sumatra, retained four out of five; Table 3; Fig. 4). The highest relative defaunation is in Southwest Asia, although the overall number of species loss is smaller than in other areas.

Discussion

This is, to our knowledge, the first attempt to produce a spatially explicit description of megafaunal loss in Asia in historical times. We found that seven of the ten species in our analyses have suffered drastic range reduction in historical times. These are shocking figures that show the dire situation of Asian megafauna and the tendency towards a neotropicalization (a term coined by Richard Corlett) of tropical Asia. Importantly, we show that megafaunal loss has occurred not only in human-dominated landscapes but also in PAs – areas explicitly devoted to the conservation of biodiversity and ecological processes. Our results show a regional-scale case of megafaunal-empty forest (Redford, 1992) and a caveat of the current system of PAs in protecting ecological processes and interactions.

Larger species—whether herbivores or carnivores—had larger original distribution ranges and have also suffered the most acute range reductions. This contrasts with the results of Ceballos & Ehrlich (2002), who found no effect of body size in the range contraction patterns of 173 mammal species across the globe. Among megaherbivores, the three rhinoceros species have suffered the most dramatic range reductions, indicating that they are an especially vulnerable clade. Rhinos have been long persecuted in Asia for the medicinal value falsely attributed to their horns (Ellis, 2006). At present, rhinos can be considered ecologically extinct sensu (McConkey & Drake, 2006) throughout most Asian ecosystems and chances are that within the next few decades Sumatran and Javan rhinos will become extinct, both in the wild and captivity. Such a tragedy would be in line with the trend in the past few decades in which several rhino taxa have been declared extinct in the wild: mainland Javan rhinos in 2010, northern white rhinos in 2010 (Ceratotherium cottoni), (Emslie, 2012), and western black rhinos Diceros bicornis longipes in 2011 (Emslie, 2012). All these taxa were driven to extinction by human persecution. Although an alternative view suggests that rhinos got disappeared from their historical range in China due to mainly climatic factors (Elvin, 2004), we do not think that climate has played an important role compared with hunting and direct human competition for good habitats during the study period.

Asian elephants, the largest of Asian terrestrial animals, have shown dramatic range contraction which according to a previous estimate is >95% by Sukumar (2006). Most of this loss occurred in southwest Asia (Turkey, Iraq, Iran, Afghanistan, and Pakistan), where elephants disappeared a long time ago (Olivier, 1978) as well as in China, where elephants have been gradually “retreating” over the past 2.5–3 thousand years until remaining isolated in a small area of Yunnan’s province (Olivier, 1978; Elvin, 2004). Elephants got extinct from Java in the 18th century (Cranbrook, Payne & Leh, 2008). In India, where approximately 60% of the remaining wild Asian elephant population occurs nowadays (Sukumar, 2006), they have also lost most of the range. In other parts of tropical Asia, the elephant range has become highly fragmented in recent times, for example, in Sumatra they have recently been declared critically endangered after losing nearly two thirds of the subspecies habitat in one elephant generation (Gopala et al., 2011). Bornean elephants are considered native now.

Asian tapirs are one of the few Asian megafaunal species that are not persecuted for Chinese Traditional Medicine (Kawanishi, Sunquist & Othman, 2002), and whose meat is not popular (especially in Malaysia, where they are considered non-halal, that is, not permissible food under the Islamic law). For these reasons, there is a general assumption that tapir populations are not under high pressure (Kawanishi, Sunquist & Othman, 2002). Our results, however, reveal a worrying situation with a dramatic reduction of 98% of their historical range and the complete disappearance from China, Laos PDR, Vietnam, Cambodia, and most of Myanmar and Thailand. Available data suggest that tapirs occur at relatively low densities, at least in Peninsular Malaysia (Rayan et al., 2012). Altogether, this depicts a more negative picture for tapir populations than often assumed.

Gaurs show the smallest range contraction among our studied megaherbivores, but this still amounts to almost three-quarters of their original range. Gaurs have probably been intensively hunted for their meat (Choudhury, 2002) throughout most of their range, to the point of being extirpated from Nepal, Bhutan, northern India, Bangladesh, sough China, and much of Indochina and the Malay Peninsula (Fig. 2B). Although the gaur was the only wild bovid included in this study, tropical Asia is home to other large and threatened wild bovids, notably the banteng (Bos javanicus; Endangered), kouprey (Bos sauveii; Critically Endangered and probably extinct), lowland anoa (Bubalus depressicornis; Endangered), mountain anoa (Bubalus quarlesi; Endangered), and the tamaraw (Bubalus mindorensis; Critically Endangered), among others. Most of these species have extremely reduced distribution ranges, often limited to island relic populations. We did not include these species in our analysis due to the difficulty to find information about their historical range.

A decline in the density of terrestrial herbivores, in turn, may threaten the largest carnivores like tigers, and the eventual loss of apex predators (trophic downgrading) leads to impacts that may cascade down through the food web. Among the four large carnivores studied, Asiatic lions lost almost all of their historical range and are now restricted to a single location in the Gir forest of India. Tigers have also got their ranges drastically reduced in history. In the last century alone, three tiger subspecies have been lost: the Caspian (P. t. virgata), Javan (P. t. sondaica), and Bali (P. t. balica) tigers, while the South China tiger (P. t. amoyensis) is probably extinct in the wild. Most of the range loss for tiger occurred in southwest Asia, Central Asia, and China. Dinerstein et al. (2006) and Walston et al. (2010) have estimated that tigers lost 93% of their range, a figure very similar to our estimate in the current study. Much of this decline occurred in the last two centuries as the result of active persecution by colonial rulers. In French Indochina, for example, as many as 45,000 tigers could have been killed between 1,860 and 1940 (Guérin, 2010). As many as 8,000 people might have been killed by tigers in Indochina during that same period (M. Guerin, 2013, personal communication).

There seems to be a strong gradient of a higher diversity of megafaunal species in mainland East and Southeast Asia that declines towards the west (Fig. 3A). Historically, in some areas of Southeast Asia such as Taman Negara (Peninsular Malaysia) and Tonle Sap Biosphere Reserve (Cambodia) more than six of these megafaunal species occurred. The Himalayan Hills and the islands of Borneo and Sumatra are also areas with particularly high levels of megafaunal presence in historical times. The loss of megafauna has been most severe in parts of Indochina, East Asia, and the Himalayan Hills, where often more than five species of megafauna are missing in the Protected Areas (Fig. 4C).

We used Tropical Asia’s network of protected areas as a proxy for healthy—or at least conservation-relevant—ecosystems. We found that more than 90% of tropical Asia’s PAs have lost one or more megafauna species. These results coincide with previous studies that point out to tropical Asia, at least Southeast Asia, as a particularly sensitive area in terms of current defaunation patterns (Ceballos & Ehrlich, 2002; Morrison et al., 2007; Ripple et al., 2016, 2017). The results of our current study on mega defaunation can be compared with those that have shown that some areas of the world still retain intact mammal assemblages. For example, Morrison et al. (2007) compared the historical range maps of large mammals with their current distribution to determine areas that have retained complete assemblages of large mammals. They have shown that some regions of the world have been successful in keeping their fauna intact, 21% of terrestrial surface all of the large mammals more than 20 kg body weight once they contained. They also showed that 12% of the total area retaining large mammal assemblages are formerly protected, the degree of protection ranging from 9% in the Palearctic to 44% in the Indo-Malayan region. However, a key question regarding the loss of megafauna from Protected Areas is whether these species have been lost in these PAs before or subsequently to the establishment of the PAs. As it is evident from the history of Protected Areas established, since Yellow Stone National Park in USA, all Protected areas have brief history (few hundred years at maximum) of establishment, therefore, we cannot establish that megafaunal loss occurred from the protected areas, because after these areas were set out as protected, much more protection was available to the megafaunal species. Therefore, it is evident that megafaunal loss from PA’s had already occurred before these areas were set out as Protected Areas.

The large reductions in local megafaunal assemblages must have significant consequences for ecosystems. For example, Corlett (2013) showed co-extinctions of parasites and co-extinctions of commensalists and mutualists because host-specific commensalists and mutualists are also vulnerable. Similarly, Campos-Arceiz & Blake (2011) showed that both African (Loxodonta spp.) and Asian elephants have unique roles as long-distance dispersal agents for seeds of all sizes, including those too large for alternate frugivores to swallow. The next largest non-ruminant mammal in much of Southeast Asia, the Asian tapir, is unlikely to disperse large seeds from large fruits (Campos-Arceiz et al., 2012).

The loss of megaherbivores releases some plant resources for surviving competitors but feeding by megaherbivores may sometimes facilitate feeding by smaller species by increased browse availability near the ground (Makhabu, Skarpe & Hytteborn, 2006). The competitive interactions between predators can be complex and unpredictable but it has been documented that loss of top carnivores as apex predators results in “trophic downgrading” (Estes et al., 2011). Megafaunal loss can also affect climate. For example, all mammalian herbivores produce methane (Franz et al., 2011) and that late Pleistocene spike in megafaunal declines resulted in a rapid loss in methane production, consequently triggering the abrupt younger dryas (12,800–11,500 B.P.) cooling event (Smith et al., 2010; Smith, Elliott & Lyons, 2011). However, carbon dioxide appears to be the primary driver of temperature changes at the end of the last glacial period (Shakun et al., 2012).

Several issues could not be properly addressed in this study. First, we could not include as many species in our study as we have liked. Ideally, we would like to have included more wild bovids (e.g., banteng), large cats (e.g., snow leopard, Uncia uncia), bears, and large primates but were constrained by the availability of historical distribution data. We hope these gaps will be filled up in future studies. Second, the reconstruction of historical ranges was based on data obtained from different sources and the amount and quality of data available were highly variable across species, ranging from 458 historical distribution points for elephants to 29 in the case of tapirs. Our historical ranges likely differ from the real ones and they are more accurate for some species than for others, and for some areas than for others (e.g., depending on the availability of fossil records). Thirdly, we had difficulty in assessing the reliability of some of our historical records, as well as in assigning geographical locations to some records that were expressed loosely. Moreover, we focus on the loss of megafauna in PAs because much of the non-protected land in tropical Asia has been severely modified and occupied by humans, making it not suitable for the presence of very large, often conflict-prone species, such as elephants and tigers. This is not to say that non-protected areas cannot or should not host megafauna, but it is more difficult to discriminate between areas that are suitable for megafauna and areas that are not. Finally, we also found difficulty in finding an appropriate metric to quantify defaunation since we used just changes in species richness without consideration of the particular species lost. Recent work in objectively quantifying defaunation (Giacomini & Galetti, 2013) is very promising and we expect more work developing in this direction.

How robust are the earlier distribution records compiled here? Some other studies published have reported some biases in this regard, for example, (Monsarrat, Boshoff & Kerley, 2019) demonstrated spatial biases in reporting historical distributions of large mammals. Also, Monsarrat & Kerley (2018) also reported taxonomic biases in the historical reporting of large mammals. Therefore, in the current study, we do understand and realize and recognize the risks inherent in such biases and obviously, these may influence the study outcomes. This is especially the case given the absolute paucity of data for some of the species focused in the current study and across such a huge area, as well as the varying socio-political histories (and hence reporting prospects) across their study area.

Throughout the world, there is an increasing interest in restoring ecological processes, including the recovery of long-missing wildlife and the ecological processes they are part of (Donlan et al., 2005). With so much range lost by Asian megafauna, conservation objectives should focus not only on protecting extant populations—the main priority—but also on restoring lost populations and the ecological role of megafauna. Our maps can be used as a tool to prioritize rewilding projects in tropical Asia. Examples of successful rewilding efforts include the reintroduction of beavers throughout much of Europe (Dewas et al., 2012) or wolves in parts of North America (Licht et al., 2010). Rewilding projects exist throughout the world but are more common in temperate latitudes (Fraser, 2009). In Tropical Asia, it seems inevitable to initiate discussions about the feasibility of range recovery of megafaunal species. Captive populations of animals like Asian elephants, tigers, or gaurs are very common and could be used as founders for rewilding. Much of tropical Asia has experienced rapid economic development in recent decades, especially China, and we think the time is ripe for the region to start seriously considering the recovery of its megafauna through rewilding projects.

Conclusion

Our study provides an insight on defaunation/range contraction of important herbivore and carnivore species and our findings can be used to guide conservation policies, especially for ecological restoration projects. Historically, the selected megafauna species were found more widely distributed than at current. By groups, rhinos showed the most dramatic range changes, followed closely by Asiatic lions, tapirs, tigers, and elephants. Defaunation was extreme in parts of East and Southeast Asia with Protected Areas having lost up to eight megafaunal species.

Supplemental Information

Supplemental Information 1 Checklist of reported items in Meta-Analysis.

Click here for additional data file.

Supplemental Information 2 Criteria for selection of studies included in meta-analysis.

Click here for additional data file.

Additional Information and Declarations

Competing Interests

Author Contributions

Data Availability

The authors declare that they have no competing interests.

Tariq Mahmood conceived and designed the experiments, performed the experiments, analyzed the data, prepared figures and/or tables, authored or reviewed drafts of the paper, and approved the final draft.

Tuong Thuy Vu conceived and designed the experiments, performed the experiments, analyzed the data, prepared figures and/or tables, authored or reviewed drafts of the paper, and approved the final draft.

Ahimsa Campos-Arceiz conceived and designed the experiments, performed the experiments, analyzed the data, prepared figures and/or tables, authored or reviewed drafts of the paper, and approved the final draft.

Faraz Akrim performed the experiments, analyzed the data, prepared figures and/or tables, authored or reviewed drafts of the paper, and approved the final draft.

Shaista Andleeb performed the experiments, analyzed the data, prepared figures and/or tables, authored or reviewed drafts of the paper, and approved the final draft.

Muhammad Farooq performed the experiments, analyzed the data, prepared figures and/or tables, authored or reviewed drafts of the paper, and approved the final draft.

Abdul Hamid performed the experiments, analyzed the data, prepared figures and/or tables, authored or reviewed drafts of the paper, and approved the final draft.

Nadeem Munawar performed the experiments, analyzed the data, prepared figures and/or tables, authored or reviewed drafts of the paper, and approved the final draft.

Muhammad Waseem performed the experiments, analyzed the data, prepared figures and/or tables, authored or reviewed drafts of the paper, and approved the final draft.

Abid Hussain performed the experiments, analyzed the data, prepared figures and/or tables, authored or reviewed drafts of the paper, and approved the final draft.

Hira Fatima performed the experiments, analyzed the data, prepared figures and/or tables, authored or reviewed drafts of the paper, and approved the final draft.

Muhammad Raza Khan performed the experiments, analyzed the data, prepared figures and/or tables, authored or reviewed drafts of the paper, and approved the final draft.

Sajid Mahmood performed the experiments, analyzed the data, prepared figures and/or tables, authored or reviewed drafts of the paper, and approved the final draft.

The following information was supplied regarding data availability:

All data is available in Tables 1–3 and the Figures.

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
