# Peer review of "Historical and current distribution ranges and loss of mega-herbivores and carnivores of Asia"

_PeerJ, doi:10.7717/peerj.10738_

## Round 0.1 · original submission · Major Revisions

Thank you very much for an interesting paper. The reviewers gave you the comments for improving and completing your article before publishing. I am very excited about receiving your revision version soon.

·

Basic reporting

The papers is well organised and clearly written. There are some minor lapses in English expression that I note in comments to the authors.

Experimental design

This is solid. The study is primarily descriptive, reporting summary data as maps. This is fine, other than for one point concerning screening for data quality that I mention in comments to authors.

Validity of the findings

The findings are sound, and I have no doubt significant enough to justify publication.

Additional comments

This is a valuable analysis, which provides strong and useful results. The paper is well organised and clearly written. I have only three substantial comments, and some minor corrections to the text.

1. at Lines 127-129 the authors comment that primary data on species locations were screened for data quality. I think some more explanation is needed on criteria that were used to wither accept or exclude records on the basis of their strength. The point is an important one as a matter of scientific rigour, and also because is is clear from comments in the Results that significant numbers of records were excluded.

2. The database the authors have collected seems very valuable. It is not clear from the review copy opf the paper whether the database will be made available in supplementary material or in some accessible data archive. In my view, it should be made available.

3. In Figure 1, it is difficult to see the current distributions. Perhaps this should be shown on separate maps without the overlays that make them difficult to see on the current maps.

Some minor corrections to the text:

Line 55: change ‘on’ to ‘in’.

Line 56: ‘OWEN-SMITH’ should not be in capitals. Same for SOULE at line 57, and in a few other cases.

Line 64: a more up-to-date reference might be valuable here. I would suggest Smith et al (2018) Science 360, 310.

Lines 183-189: there is some repetition here, that should be removed.

Line 300: the word ‘less’ should be inside the brackets.

Line 334-335: can you give a reference and brief explanation for this term? I’m not sure what it means.

Line 352: should be ‘persecution’ rather than ‘prosecution’.

Line 353: ‘got vanished’ should be changed to something like ‘disappeared from’.

Line 360: should be ‘retreating’ rather than ‘retrieving’, I think.

·

Basic reporting

The MS is general well written, although there are some minor technical problems that need addressing, such as the inconsistent use of capitals when citing references. Avoid the word “got” and see lines 421-422 for a confusing sentence.

Title: This currently too broad and ambitious as it suggests that all megafauna are covered in all areas of Asia, whereas this study is restricted to a subset of ten species and not all of Asia is covered. Interestingly, the offered short title is more evocative, although it again needs refining and the gardening term is not valid based on data provided here.

The section on other studies of range contractions (lines 77-94) does not cover the important studies in Asia by Turvey and his collaborators, nor the comprehensive historical mapping of large mammals in China by Wen and his collaborators. A useful recent paper is Qian et al (2020. Changes in the Historical and Current Habitat Ranges of Rare Wild Mammals in China: A Case Study of Six Taxa of Small- to Large-Sized Mammals. Sustainability 12, 2744; doi:10.3390/su12072744), which covers some of the species addressed here for the area of China, and also highlights considerable literature not used by Mahmood et al.

Experimental design

Mahmood et al collate information on earlier distributions of a suite of ten large mammal species and compare this to their occurrence in a set of protected areas, this in tropical/warm temperate Asia. Given the decline of large mammals globally, this is clearly a relevant and important topic, as we need to better understand the patterns, causes and consequences of these declines.

The study focuses on a set of ten large mammal species, but it is not clear what the basis for selecting these species may be. So the authors should justify this choice, this based on their hypotheses or research objectives, or some other justification around data quality, etc.

A key aspect of this study is the assessment of the representation of these taxa in a set of the larger protected areas - this is an important focus as it talks to the efficacy of conservation areas and should possibly be highlighted in the title.

How robust are the earlier distribution records compiled here? Monsarrat et al (2018. Ecography 41:1-12. https://doi.org/10.1111/ecog.03944) recently demonstrated spatial biases in reporting of historical distributions of large mammals. In addition, Monsarrat & Kerley (2018. Biological Conservation 223:68-75. https://doi.org/10.1016/j.biocon.2018.04.036) demonstrated taxonomic biases in the historical reporting of large mammals. The authors need to recognise the risks inherent in such biases and how these may influence their study. This is especially the case given the absolute paucity of data for some of their species across such a huge area, as well as the varying socio-political histories (and hence reporting prospects) across their study area.

The delimitation of the study period as “a period of several thousand years” (line 96) is rather vague and also needs to be justified. Later this is given as ~10 000 years (line 110) – which is more specific but also not equivalent to “several”.

The method of developing historical occurrence maps is extremely crude, given that it is based on interpolation between location records, as well as some apparently arbitrary (as seen in the maps provided) extrapolation beyond these records. Hence, this approach does not reflect any assessment of habitat suitability in these ranges.

The classification of occurrence in the various PAs (lines 158-159) assumes that the habitat within the PA was, and still is, currently suitable for the species in question. This needs to be validated.

The estimation of proportional loss of megafuanal species from PAs (lines 163-169) will provide a misleading answer, given that the analyses focusses only on a subset of the megafauna that may have occurred in these PAs. Hence this section of the study is not valid and should be removed.

The section on the literature used (lines 173- 180) adds no value to this study as this is background information and does not provide information on the actual data used. The authors should provide these datasets of literature used and their validity in a supplementary information file or accessible database, and delete this section. This should be replaced with a description of the number of actual records and their representation through time for each species. These data should also be made available in a data repository - see submission system.

Lines 297-303 are to some extent a repetition of the section in the Methods. Delete to focus on Results.

Validity of the findings

The historical data analysed for each species ranges from 29 to 458 points (no data provided for leopard), and this for a period of up to 10 000 years. There is clearly a need to justify the extrapolation of these sparse data over millions of km2 and thousands of years. In addition, there is a need to validate the application of the range estimation approaches using such sparse data. Clearly, simply filling the gaps between points (as done here) has considerable risk of overestimating areas occupied, particularly when there is no assessment of habitat suitability in the intervening areas. This is glossed over in the Discussion, but is a foundational issue for this paper, and must be addressed in a robust fashion.

The findings for the 10 species reported on here cannot be used to refute (lines 340-343) a much larger earlier study based on 173 species, as the present study uses a non-random sample.
The Discussion tends to focus on detailed patterns of loss (often without supporting literature) that is at odds with the large scale of this study. This focus needs to be kept to the larger scale to be valid.
The generalization around megafuanal biogeographic patterns (starting line 404) are not justified, as only 10 taxa were considered, and this is clearly a biased sample.
The section on the loss of the ecological function of these species (lines 433-443) could be replaced by two sentences, an also needs to relate to the predators.

Additional comments

Missing from the Discussion is the concept of relaxation – and a key question would be whether these species have been lost in these PAs prior to or subsequently to the establishment of the PAs.

---

## Round 0.2 · accepted · Accept

Congratulations, now your manuscript is ready to publish.

·

Basic reporting

no comment

Experimental design

no comment

Validity of the findings

no comment

Additional comments

I am happy with the revised version of the manuscript. My one further comment relates to availability of primary data and sources as supplementary material. In their response the authors say that these data and sources can be made available if the journal editor requests it. It is my strong view that the data should be provided in this way.